# Deacetylasperulosidic Acid Ameliorates Pruritus, Immune Imbalance, and Skin Barrier Dysfunction in 2,4-Dinitrochlorobenzene-Induced Atopic Dermatitis NC/Nga Mice

**DOI:** 10.3390/ijms23010226

**Published:** 2021-12-25

**Authors:** Jin-Su Oh, Geum-Su Seong, Yong-Deok Kim, Se-Young Choung

**Affiliations:** 1Department of Life and Nanopharmaceutical Sciences, Graduate School, Kyung Hee University, 26 Kyungheedae-ro, Dongdaemun-gu, Seoul 02447, Korea; ok9638@naver.com; 2Korea Food Research Institute, 245 Nongsaengmyeong-ro, Iseo-myeon, Wanju-gun 55365, Korea; gsseong824@nate.com; 3NST BIO Co., Ltd., Goeumdal-ro, Yangchon-eup, Gimpo-si 10049, Korea; ydkim@nstbio.co.kr; 4Department of Preventive Pharmacy and Toxicology, College of Pharmacy, Kyung Hee University, 26 Kyungheedae-ro, Dongdaemun-gu, Seoul 02447, Korea

**Keywords:** atopic dermatitis, fermented *Morinda citrifolia*, deacetylasperulosidic acid, NC/Nga mice, pruritus, immune balance, skin barrier function

## Abstract

The prevalence of atopic dermatitis (AD), a disease characterized by severe pruritus, immune imbalance, and skin barrier dysfunction, is rapidly increasing worldwide. Deacetylasperulosidic acid (DAA) has anti-atopic activity in the three main cell types associated with AD: keratinocytes, mast cells, and eosinophils. Our study investigated the anti-atopic activity of DAA in 2,4-dinitrochlorobenzene-induced NC/Nga mice. DAA alleviated the symptoms of AD, including infiltration of inflammatory cells (mast cells and eosinophils), epidermal thickness, ear thickness, and scratching behavior. Furthermore, DAA reduced serum IgE, histamine, and IgG1/IgG2a ratio and modulated the levels of AD-related cytokines and chemokines, namely interleukin (IL)-1β, IL-4, IL-6, IL-9, IL-10, IL-12, tumor necrosis factor-α, interferon-γ, thymic stromal lymphopoietin, thymus and activation-regulated chemokine, macrophage-derived chemokine, and regulated on activation the normal T cell expressed and secreted in the serum. DAA restored immune balance by regulating gene expression and secretion of Th1-, Th2-, Th9-, Th17-, and Th22-mediated inflammatory factors in the dorsal skin and splenocytes and restored skin barrier function by increasing the expression of the pro-filaggrin gene and barrier-related proteins filaggrin, involucrin, and loricrin. These results suggest DAA as a potential therapeutic agent that can alleviate the symptoms of AD by reducing pruritus, modulating immune imbalance, and restoring skin barrier function.

## 1. Introduction

Atopic dermatitis (AD) is a chronic recurrent skin disease characterized by severe pruritus, immune imbalance, and defective skin barrier function [1]. The main hallmark of AD is the biased differentiation of T helper (Th)2 cells [2]. A biased Th2-mediated immune response causes Th1/Th2 immune imbalance and impairs skin barrier function [3,4,5]. Furthermore, these phenotypes lead to clinical symptoms, such as erythema, maceration, dry skin, abrasions, and lichenification [6,7,8,9].

In patients with AD, the expression and secretion of the AD-related cytokines interleukin (IL)-25, IL-33, thymic stromal lymphopoietin (TSLP), and the chemokines thymus and activation-regulated chemokine (TARC), macrophage-derived chemokines (MDC) and, regulated on activation, normal T cell expressed and secreted (RANTES) are increased in keratinocytes [10,11]. The cytokines IL-25, IL-33, and TSLP can stimulate Th2 cells directly or indirectly by stimulating several immune cells, including dendritic cells, mast cells, and eosinophils [3,12]. Furthermore, TARC and MDC bind to C–C motif chemokine receptor 4 (CCR4) expressed on the surface of Th2 cells, thereby stimulating the activation and infiltration of Th2 cells into skin lesions [13,14,15]. RANTES is involved in chronic AD by promoting the degranulation and infiltration of eosinophils [16].

Activated Th2 cells express and secrete the cytokines IL-4, IL-5, IL-13, and IL-31 [17]. IL-4 and IL-13 increase immunoglobulin E (IgE) and IgG1 production through isotype switching in B cells [18]. Histamine is a biological response modulator that is released from degranulated mast cells, causing itching [19,20]. IL-4 increases the activity of Th2 cells and inhibits the activity of Th1 cells [21]. Consequently, Th1-mediated cytokines, such as interferon-γ (IFN-γ) are decreased, resulting in a Th1/Th2 immune imbalance. IL-5 and IL-31 induce chronic AD by increasing the number of eosinophils by inhibiting apoptosis [22,23]. Moreover, IL-31 impairs skin barrier function by increasing pruritus [23,24].

Th2-mediated cytokines can additionally activate Th9, Th17, and Th22 cells [25,26]. Th9, Th17, and Th22 cells have been reported to cause defects in skin barrier function [7,26]. Th9 cells are activated by the Th2-mediated cytokines IL-4 and IL-10 [26,27,28]. Th9 cells produce IL-9 and IL-10, which trigger IgE production in B cells and stimulate keratinocytes [26,29]. IL-9 and IL-10 are also expressed and secreted by activated Th2 cells [30,31]. IL-17 and IL-22 are produced by Th17 cells [32], and IL-22 is produced specifically by Th22 cells [33]. IL-17 and IL-22 cause defects in skin barrier function-related proteins, including filaggrin (FLG), involucrin (IVL), and loricrin (LOR) [34].

The skin barrier protects the skin by blocking the influx of external antigens and allergens and maintaining water homeostasis [35]. FLG, IVL, and LOR proteins are key components of skin barrier function [7,36]. FLG exists in the form of pro-FLG and is converted to FLG through dephosphorylation by serine proteases [37]. FLG is transformed into free amino acids and then converted into components of several natural moisturizing factors [38]. IVL and LOR promote the final differentiation of the keratinocytes [39]. Several AD-related cytokines and chemokines, including the Th2 cytokines IL-4, IL-13, and IL-31, cause defects in skin barrier function [26,36,39,40]. Therefore, the improvement of pruritus and restoration of immune balance and skin barrier function are key to AD treatment.

Current medications for AD include corticosteroids (TCS). However, TCS have side effects, such as skin rash and hypertension [41]. Therefore, AD therapeutic agents with fewer side effects should be developed.

The NC/Nga mouse is a widely used animal model to study the efficacy and mechanisms associated with AD [42,43]. Repeated application of 2,4-dinitrochlorobenzene (DNCB) induces AD-like skin lesions through elevated IgE and Th2 immune responses [44].

Deacetylasperulosidic acid (DAA) is an iridoid component present at high concentration in Noni (*Morinda citrifolia*), and its content is increased by fermentation [6,45]. In a previous study, DAA was selected as a functional ingredient and its anti-atopic efficacy was demonstrated in the three main cell types related to AD: keratinocytes, mast cells, and eosinophils [45]. Therefore, this study investigated the mechanism by which DAA alleviates AD-related clinical symptoms and restores Th1/Th2 immune balance and skin barrier function using a DNCB-induced NC/Nga mouse model.

## 2. Results

### 2.1. Deacetylasperulosidic Acid (DAA) Attenuates 2,4-Dinitrochlorobenzene (DNCB)-Induced Atopic Dermatitis (AD)-like Symptoms, Ear Thickness, and Scratching Behavior

Figure 1A shows representative images of the six test groups, including the normal, control, predinisolone (PD), and 3, 6, and 12 mg/kg DAA groups during weeks 10–14 after induction of AD. Repetitive application of DNCB aggravated the clinical symptoms associated with AD, including erythema, maceration, dryness, abrasions, lichenification, scratching behavior, and ear thickness, over the first nine weeks. The dermatitis score, scratching behavior, and ear thickness were significantly higher in the control group than in the normal group. DAA administration significantly reduced these parameters in a dose-dependent manner (Figure 1B–D), resulting in similar efficacy in the 3 mg/kg DAA and PD groups on week 14.

### 2.2. DAA Reduces the Thickening of the Epidermis, along with Eosinophil and Mast Cell Infiltration

Epidermal thickness and the number of infiltrated inflammatory cells were investigated to confirm the efficacy of DAA in the dorsal skin of NC/Nga mice. Epidermal thickness and the number of infiltrated inflammatory cells were significantly higher in the control group than in the normal group. DAA treatment reduced the epidermal thickness and the number of infiltrated inflammatory cells in a dose-dependent manner, resulting in similar efficacy between the 3 mg/kg DAA and PD groups (Figure 2).

### 2.3. DAA Reduces Immunoglobulin E (IgE) and Histamine Levels in the Serum of NC/Nga Mice

IgE and histamine levels were significantly higher in the control group treated with DNCB than in the normal group. DAA decreased serum IgE and histamine levels in a dose-dependent manner. DAA exhibited a higher inhibitory effect on IgE and histamine than PD (Figure 3).

### 2.4. DAA Restores the IgG1/IgG2a Balance in the Serum

The restoration of Th1/Th2 balance was investigated by measuring the levels of serum IgG1 and IgG2a on DAA treatment. The serum IgG1 level of the DNCB-induced control group was significantly higher than that of the normal group. DAA reduced serum IgG1 levels in a dose-dependent manner, with similar efficacy between the 3 mg/kg DAA and PD groups (Figure 4A). Furthermore, the serum IgG2a level of DNCB-induced mice was significantly higher than that of the normal mice. When compared to the control group, the DAA group displayed a dose-dependent increase in IgG2a levels, whereas IgG2a levels decreased in the PD group (Figure 4B). DAA reduced IgG1 and increased IgG2a levels; therefore, the level of IgG1/IgG2a was normalized in a dose-dependent manner. In the PD group, both IgG1 and IgG2a levels were reduced, resulting in a decrease in the IgG1/IgG2a ratio (Figure 4A–C).

### 2.5. DAA Regulates Serum Levels of AD-Related Cytokines and Chemokines

The levels of the cytokines IL-4, IL-9, IL-10, and TSLP secreted by activated Th2 cells, and the levels of IL-1β, IL-6, and tumor necrosis factor-α (TNF-α) involved in acute and chronic AD were significantly higher in the control group than in the normal group. Furthermore, the levels of the chemokines TARC, MDC, and RANTES, which directly or indirectly activate Th2 cells, were significantly increased in the control group compared to levels in the normal group. DAA decreased the levels of cytokines and chemokines in a dose-dependent manner, resulting in a similar efficacy between the 3 mg/kg DAA and PD groups (Figure 5A–C). Th1-mediated cytokines IL-12 and IFN-γ were significantly decreased in the control group compared to levels in the normal group. DAA restored serum IL-12 and IFN-γ levels in a dose-dependent manner compared to levels in the control group. We found that 12 mg/kg DAA restored the level of IL-12 and IFN-γ to that observed in the normal group (Figure 5D). DAA reduced Th2 and Th9-related cytokines and improved AD by increasing Th1-mediated cytokines. However, PD reduced Th1-, Th2-, and Th9-mediated cytokines and chemokines.

### 2.6. DAA Regulates the Secretion Levels of AD-Related Cytokines in Splenocytes

The levels of Th2 cytokines IL-4, IL-5, IL-10, IL-13, IL-31, IL-33, and TSLP were significantly higher in the control group than in the normal group. Moreover, the levels of acute and chronic AD-related IL-1β, IL-6, TNF-α, TARC, MDC, and RANTES were significantly higher in the control group than in the normal group. Furthermore, the levels of Th9-, Th17-, and Th22-related cytokines IL-9, IL-17, and IL-22 were significantly higher in the control group than in the normal group. DAA reduced the levels of AD-related cytokines and chemokines in a dose-dependent manner, resulting in a similar efficacy between the 3 mg/kg DAA and PD groups (Figure 6A–D). Th1-mediated cytokines IL-12 and IFN-γ were significantly decreased in the control group compared to levels in the normal group. DAA restored the secretion levels of IL-12 and IFN-γ in a dose-dependent manner compared to levels in the control group. We found that 12 mg/kg DAA restored the secretion level of IL-12 and IFN-γ to that observed in the normal group (Figure 6E). DAA modulated the balance of AD-related cytokine and chemokine production in splenocytes, while PD decreased Th1, Th2, Th9, Th17, and Th22-mediated cytokine and chemokine production.

### 2.7. DAA Inhibits the Gene Expression of Th2 Cell Activation-Related Cytokines, Chemokines and CCR4 in the Dorsal Skin of Mice

The gene expression level of CCR4, a receptor for TARC and MDC, was significantly higher in the control group than in the normal group. Furthermore, the levels of IL-25, IL-33, TSLP, TARC, MDC, and RANTES, which directly or indirectly activate Th2 cells in the dorsal skin of mice, were significantly increased in the control group compared to levels in the normal group. DAA reduced the expression levels of CCR4, cytokines, and chemokines in a dose-dependent manner, resulting in a similar efficacy between the 3 and 6 mg/kg DAA and PD groups (Figure 7).

### 2.8. DAA Regulates the Gene Expression of AD-Related Cytokines and Chemokines in the Dorsal Skin of NC/Nga Mice

The gene expression of Th2-mediated cytokines IL-4, IL-5, IL-10, IL-13, and IL-31 was significantly higher in the control group than in the normal group. Gene expression of the acute and chronic AD-related IL-1β, IL-6, and TNF-α as well as Th9-, Th17-, Th22-related IL-9, IL-17, and IL-22, respectively, was significantly increased in the control group compared to levels in the normal group. DAA treatment reduced the gene expression of AD-related cytokines and chemokines, including Th2-mediated cytokines, in a dose-dependent manner, resulting in a similar efficacy between the 3 mg/kg DAA and the PD groups (Figure 8A–C). Th1-mediated cytokines IL-12 and IFN-γ were significantly decreased in the control group compared to levels in the normal group. DAA restored the gene expression levels of IL-12 and IFN-γ in a dose-dependent manner compared to levels in the control group. We found that 12 mg/kg DAA restored the gene expression level of IL-12 and IFN-γ to that observed in the normal group (Figure 8D). DAA treatment attenuated immune imbalance by modulating AD-related cytokine and chemokine gene expression in the dorsal skin, while PD treatment decreased AD-related cytokine and chemokine gene expression.

### 2.9. DAA Alleviates the Defects of the Skin Barrier Function

The expression of skin barrier-related FLG, IVL, and LOR proteins was significantly lower in the DNCB-induced control group than in the normal group. DAA treatment increased the abundance of proteins in a dose-dependent manner compared to that in the control group (Figure 9A,C). The gene expression of pro-FLG was significantly decreased in the DNCB-induced control group compared to that in the normal group. DAA treatment increased pro-FLG gene expression in a dose-dependent manner (Figure 9B). We found that 12 mg/kg DAA restored the expression of skin barrier-related proteins and pro-FLG gene to a level similar to that in the normal group. However, PD slightly restored the expression of the skin barrier proteins FLG, IVL, and LOR when compared to DAA (Figure 9A,C).

## 3. Discussion

In the present study, the efficacy of DAA against AD was investigated using an in vivo animal model, with a particular focus on the mechanism by which DAA alleviates AD-related clinical symptoms, scratching behavior, immune imbalance, and recuperation of skin barrier function. Pruritus is an important clinical feature of AD and is one of the main diagnostic criteria. Controlling the “itch–scratch” cycle is important in the treatment of AD and is associated with a reduction in clinical symptoms, immune cell infiltration, ear thickness, and epidermal thickening [46]. DAA treatment successfully improved AD-related clinical symptoms, ear thickness, and scratching behavior (Figure 1). Moreover, histological analysis showed a decrease in epidermal thickness and the number of infiltrated inflammatory cells (Figure 2). These results suggest that DAA may be clinically useful in AD treatment. Furthermore, serum IgE and histamine levels decreased after DAA treatment, indicating that DAA was effective in reducing pruritus by reducing the levels of IgE and histamine, which are involved in itching (Figure 3).

Antigen penetration through the defective epidermal barrier activates keratinocytes, mast cells, and eosinophils, which secrete cytokines, including IL-4 and IL-13, to increase Th2 cell activity [47,48,49,50]. DAA treatment reduced the gene expression and secretion of Th2 cytokines, including IL-4, IL-5, IL-13, and IL-31, in the dorsal skin and splenocytes, thereby reducing Th2-mediated cytokine levels in serum (Figure 5A, Figure 6A and Figure 8A). The decrease in IL-4 and IL-13 levels was consistent with the decrease in serum IgG1 levels (Figure 4A). Th1-mediated cytokines IL-12 and IFN-γ are associated with the activity of Th1 cells [51]. IL-12 activates Th1 cells, and IFN-γ secreted from Th1 cells reduces Th2 cell activity and IL-4 secretion, thereby suppressing IgE and IgG1 production and inducing IgG2a production in B cells [52,53]. DAA treatment increased the levels of the Th1-mediated cytokines IL-12 and IFN-γ in the serum, splenocytes, and dorsal skin (Figure 5D, Figure 6E and Figure 8D). Thus, DAA increased the level of serum IgG2a by decreasing IL-4 and IL-13 and increasing Th1-mediated cytokines (Figure 4B). Consequently, DAA treatment normalized the IgG1/IgG2a ratio in a dose-dependent manner (Figure 4C). In the PD group, the IgG1/IgG2a ratio was restored by the reduction of IgG1 and IgG2a. Collectively, our results suggest that PD restored the immune balance by inhibiting both Th1 and Th2 cell activity, whereas DAA restored the immune balance by regulating Th1 and Th2 cell activity. Reduction of the Th2-mediated cytokines IL-5 and IL-31 suggests that DAA reduced eosinophil infiltration, differentiation, and survival, and inhibited the progression of chronic AD. Furthermore, we suggest that DAA may alleviate clinical symptoms, particularly pruritus, through the reduction of IL-31.

The AD-related cytokines, namely IL-1β, IL-6, and TNF-α, are expressed and secreted by inflammatory cells such as keratinocytes, Th2 cells, mast cells, and eosinophils. They are involved in the progression of acute and chronic AD and promote the activation of Th9 [54], Th17, and Th22 cells [55,56,57]. DAA treatment reduced the gene expression and secretion levels of IL-1β, IL-6, and TNF-α in the serum, splenocytes, and dorsal skin (Figure 5B, Figure 6B and Figure 8B). These results demonstrate that DAA alleviated AD by reducing cytokine levels and inhibiting the activities of Th9, Th17, and Th22 cells involved in acute and chronic AD.

Th9-mediated IL-9 promotes the expression and secretion of IL-4 in several inflammatory cells, such as keratinocytes and mast cells [26,28]. Furthermore, IL-9 promotes mast cell and eosinophil infiltration [26,29,58]. In this study, DAA treatment reduced the levels of IL-9 in the serum, splenocytes, and dorsal skin (Figure 5A, Figure 6C and Figure 8C), indicating that DAA inhibited the activation of Th9 cells, thereby reducing the cytokine IL-4 associated with Th2 cells. This is consistent with the decrease in the number of infiltrating mast cells and eosinophils (Figure 2). Th2-mediated cytokines IL-4 and IL-10 promote the production of IL-9 in Th9 cells [27]. Our study suggests that DAA reduced both the gene expression and secretion levels of IL-4 and IL-10, thereby suppressing Th9 cell activity. Th1-mediated IFN-γ levels have been reported to be inversely related to IL-9 levels [59]. Our study showed a decrease in IFN-γ levels and an increase in IL-9 levels in the control group, while DAA treatment decreased IL-9 levels and increased IFN-γ levels (Figure 5, Figure 6 and Figure 8). These results suggest that DAA restores immune balance by decreasing IL-9 levels and increasing IFN-γ levels.

IL-17 and IL-22 are mainly related to the activation of Th2 cells and impair skin barrier function [32]. DAA treatment reduced the secretion of IL-17 and IL-22 from splenocytes and dorsal skin (Figure 6C and Figure 8C), suggesting that DAA restores immune balance by inhibiting the activity of Th2 cells and suppressing the activity of Th17 and Th22 cells. In addition, DAA restored the expression of skin barrier-related proteins in the dorsal skin (Figure 9).

Keratinocytes induce a biased Th2 immune response, which further exacerbates immune imbalance and pruritus in patients with AD [60]. TARC, MDC, and RANTES are secreted by keratinocytes, mast cells, and eosinophils, and may be involved in the chronicization of AD [11,16]. DAA reduced the expression of TARC, MDC, RANTES, and CCR4 (receptor for TARC and MDC) in serum, splenocytes, and dorsal skin (Figure 5C, Figure 6D and Figure 7). This indicates that DAA can reduce the binding of TARC and MDC by reducing CCR4 expression. Recently, IL-17 and IL-22 have been reported to increase CCR4 expression [32,61,62]. Our results demonstrated that a reduction in IL-17 and IL-22 levels decreased CCR4 expression. Collectively, the reduction of TARC, MDC, and CCR4 suppresses the activity of Th2 cells, indicating that it is effective in restoring immune balance. Reduction of RANTES reduced the growth and differentiation of eosinophils, thereby alleviating the symptoms of chronic AD and epidermal thickening [16,63,64].

IL-33 and TSLP secreted primarily from keratinocytes are increased in AD and stimulate Th2 cells, mast cells, and eosinophils [65,66]. IL-25 is mainly secreted by keratinocytes and promotes the production of Th2-mediated cytokines IL-4 and IL-5 in mast cells [67]. DAA reduced the gene expression and secretion of these cytokines in the serum, splenocytes, and dorsal skin (Figure 5A, Figure 6A and Figure 7B). These results correlate with the decrease in infiltrated mast cells and eosinophils (Figure 2).

Cytokines and chemokines produced by several immune cells, including Th2, Th9, Th17, and Th22, reduce the expression of the skin barrier-related proteins FLG, IVL, and LOR [5,6,26,34,68,69]. DAA increased the gene expression of pro-FLG and the expression of skin barrier-related proteins. In contrast, PD treatment resulted in a slight increase in skin barrier-related proteins compared to that in the control group (Figure 9). Previous studies have reported that TCS increases the skin barrier related proteins FLG and LOR [70]. PD is thought to restore skin barrier-related proteins through the reduction of AD-related cytokine levels. DAA showed a higher efficacy than PD treatment for both gene expression of pro-FLG and restoration of skin barrier-related proteins. Unlike PD, DAA treatment might alleviate AD by restoring the beneficial immune balance and skin barrier function. Overall, we suggest that DAA may be a potential anti-AD drug candidate with higher efficacy and lower toxicity than PD.

## 4. Materials and Methods

### 4.1. Reagents

DAA was provided by Phytolab (Vestenbergsgreuth, Germany). Roswell Park Memorial Institute-1640 (RPMI-1640) was purchased from Gibco BRL (Grand Island, NY, USA). Fetal bovine serum (FBS) was purchased from Gemini Bio (West Sacramento, CA, USA). Penicillin–streptomycin, DNCB, PD, and concanavalin A (Con-A) were purchased from Sigma-Aldrich (St. Louis, MO, USA). IgE was purchased from Shibayagi (Shibukawa, Japan). The cell strainer was procured from BD Biosciences (Franklin Lakes, NJ, USA). IgG1, IgG2a, IL-4, and IL-6 were purchased from Enzo Life Sciences Inc. (Farmingdale, NY, USA). Histamine, IL-5, IL-12, IL-13, IL-17, IL-22, IL-31, IL-33, TSLP, and TNF-α were purchased from Elabscience (Huston, ID, USA), and IL-9 was purchased from Abcam Inc. (Waltham, MA, USA). IL-1β, IL-10, TARC, MDC, and RANTES were purchased from R&D Systems Inc. (Minneapolis, MN, USA). IL-12 and IFN-γ were purchased from Abbkine Scientific Co. Ltd. (Wuhan, China). The RNA-spin total RNA extraction kit was obtained from INtRon Biotechnology (Seoul, Korea). Oligo dT primers were purchased from Bionics Inc. (Seoul, Korea). The cDNA synthesis kit, SYBR Green, and Rox dyes were purchased from Takara Korea Biomedical Inc. (Shiga, Japan). Lysis buffer containing cOmplete™ protease inhibitor cocktail tablets was purchased from Roche Diagnostics Ltd. (Indianapolis, IN, USA). Sodium dodecyl sulfate–polyacrylamide gel (SDS–PAGE) was purchased from Bio-Rad (Hercules, CA, USA). The Pierce™ BCA protein assay kit was purchased from Thermo Fisher Scientific Inc. (Rockford, IL, USA). Anti-β-actin, anti-filaggrin, anti-involucrin, and anti-loricrin antibodies were obtained from Santa Cruz Biotechnology Inc. (Dallas, TX, USA). Horseradish peroxidase-conjugated (HRP) secondary antibodies were obtained from GeneTex Inc. (Irvine, CA, USA).

### 4.2. Animals

Four-week-old male NC/Nga mice were purchased from Shizuoka Laboratory Center Inc. (Shizuoka, Japan). Mice were maintained on a 12 h light/dark cycle at 25 ± 5 °C and 50% ± 5% humidity in individually ventilated cages in a specific pathogen-free facility at Central Laboratory Animal and had ad libitum access to food and water (Catalog number 5L79, Central Laboratory Animal, Seoul, Korea). Serum collection was performed according to methods described in previous studies [6,7]. Briefly, on the last day of the experiment, mice were anesthetized with isoflurane (2–2.5%), and blood was immediately collected from the venous vessels, followed by euthanization using CO_2_. Blood was stored at room temperature for 1 h and centrifuged at 3000× *g* at 4 °C for 15 min to supernatant of serum. Serum samples were stored at −80 °C until use. All experimental procedures were performed according to the protocol approved by the Kyung Hee University Animal Care and Use Committee guidelines (approval number KHSASP-21-088, date of approval: 23 February 2021).

### 4.3. Application of AD-Like Skin Lesions and Treatment of DAA

DNCB application and DAA treatment were performed through the methods of previous studies [6,7]. Briefly, after acclimatization for 1 week, hair was removed from the dorsal skin of NC/Nga mice using an electric razor. Mice were randomly divided into normal (DNCB untreated group; naïve control), control (DNCB-treated group; negative control), PD (prednisolone; positive control), DAA 3, 6, and 12 mg/kg groups, and six mice were assigned to each group. To induce AD-like skin lesions, 1% DNCB was dissolved in a mixture of acetone and ethanol (2:3) and applied twice to the dorsal skin (200 μL) and right ear (100 μL) of the mice. After sensitization, 0.4% DNCB was dissolved in a mixture of acetone and olive oil (3:1) and applied repeatedly to the skin on the back (150 μL) and right ear (50 μL) thrice a week for 14 weeks. At week 10, mice in the normal and control groups were orally administered 0.5% carboxymethyl cellulose (CMC) daily. Additionally, 3 mg/kg PD and 3, 6, or 12 mg/kg DAA were each dissolved in CMC and orally administered daily for four weeks. The schedule for DNCB induction and oral administration of DAA is shown in Figure 10.

### 4.4. Dermatitis Score and Ear Thickness

Dermatitis scores were recorded three times per week based on previous studies [71]. Clinical symptoms were evaluated with a score of 0 (none), 1 (mild), 2 (moderate), and 3 (severe) for each of the five symptoms (clinical symptoms: erythema, maceration, dryness, abrasions, and lichenification). The total dermatitis score was quantified as the average of all individual scores for the five symptoms. Ear thickness was measured thrice a week in the right ear of each mouse using an ear thickness gauge (Mitutoyo Corporation, Tokyo, Japan).

### 4.5. Scratching Behavior

The scratching behavior of NC/Nga mice was recorded three times per week [72]. Briefly, the mice were acclimatized to acrylic cages for 1 h after vehicle administration. Scratching behavior with the hind paw was then measured and recorded on the skin on the neck, ears, and back for 30 min. Scores ranging from 0–4 (i.e., 0 points (none), 2 points (less than 1.5 s), 4 points (1.5 s or more) were assigned. The total score for scratching behavior was determined as the average of the individual measures.

### 4.6. Histological Analysis

Histological analysis was performed according to previously described methods [6,7,68]. Briefly, mice were sacrificed, and the dorsal skin was fixed with 10% formalin. Tissues were sectioned to a thickness of 4 μm and stained with hematoxylin and eosin and toluidine blue. After staining, the sections were observed under an optical microscope (400×, DP Controller Software, Olympus Optical, Tokyo, Japan). The epidermis thickness and the number of infiltrated inflammatory cells (e.g., mast cells and eosinophils) were measured at six sites per mouse using Image J software (National Institute of Health, Starkville, MD, USA).

### 4.7. Serum Immunoglobulin and Histamine Assay

The levels of IgE, histamine, IgG1, and IgG2a in the serum were measured using a mouse enzyme-linked immunosorbent assay (ELISA) kit. The levels of all cytokines and chemokines were measured using a mouse ELISA kit according to the manufacturer’s instructions.

### 4.8. Serum Cytokines and Chemokines Assay

The levels of the cytokines and chemokines IL-1β, IL-4, IL-6, IL-9, IL-10, IL-12, TSLP, TNF-α, IFN-γ, TARC, MDC, and RANTES in the serum were measured using a mouse ELISA kit according to the manufacturer’s instructions.

### 4.9. Splenocyte Supernatant Isolation and Cytokine and Chemokine Analysis

Splenocytes were crushed and examined as described previously [6,7,68]. Briefly, splenocytes were disrupted with the back of a sterile syringe plunger and harvested using a cell strainer. Splenocytes were treated with red blood cell lysis buffer. Thereafter, the splenocytes were washed three times with RPMI-1640 medium supplemented with 10% FBS. The isolated splenocytes were treated with 5 μg/mL Con-A, seeded in 24-well plates at a concentration of 1 × 10^6^ cells/well, and incubated for 72 h at 37 °C with 5% CO_2_. Supernatants were collected, and splenocytes were homogenized in lysis buffer containing cOmplete™ protease inhibitor cocktail tablets. The lysate was centrifuged at 4 °C and 10,000× *g* for 10 min. After centrifugation, the splenocyte supernatant and lysate were frozen at −80 °C for subsequent cytokine and chemokine analyses. Cytokines IL-1β, IL-4, IL-5, IL-6, IL-9, IL-10, IL-12, IL-13, IL-17, IL-22, IL-31, IL-33, TSLP, TNF-α, IFN-γ, TARC, MDC, and RANTES present in the supernatant were measured using an ELISA kit, according to the manufacturer’s instructions. The lysate protein concentration was measured using the Pierce™ BCA protein assay kit. The levels of cytokines and chemokines in the supernatants were normalized to the protein concentration of the lysate.

### 4.10. RNA Extraction and Quantitative Real Time-Polymerase Chain Reaction (RT-qPCR)

RNA extraction and cDNA synthesis were performed as described previously [7]. Briefly, total RNA was extracted from the dorsal skin of mice using the Easy-Red total RNA extraction kit. Chloroform was added to the extract and stored at room temperature (25 ± 5 °C) for 20 min. The supernatant was collected by centrifugation for 15 min at 10,000× *g* at 4 °C, and an equal volume of isopropanol was added to the supernatant and stored overnight. After 24 h, centrifugation was performed for 15 min at 10,000× *g* at 4 °C and washed with 75% ethanol. After removing the ethanol, the extract was dried at room temperature (25 ± 5 °C) for 1 h and then dissolved in diethyl pyrocarbonate water. Complementary DNA was synthesized using a cDNA synthesis kit and RT-qPCR was performed on an ABI StepOnePlus™ real-time PCR system (Applied Biosystems, Waltham, MA, USA) using the synthesized cDNA and SYBR Premix EX Taq. The primer sequences are listed in Table 1. The mRNA expression level was normalized to that of GAPDH using the 2^−ΔΔCt^ method to obtain the cycle threshold (Ct) value.

### 4.11. Protein Extraction and Western Blotting

The dorsal skin was isolated, and examinations were performed as previously described [6,7]. Briefly, the dorsal skin tissues frozen in liquid nitrogen were crushed using a pestle. Subsequently, the skin tissues were homogenized in lysis buffer containing cOmplete™ protease inhibitor cocktail tablets. The lysates of dorsal skin tissues were sonicated and centrifuged at 10,000× *g* for 15 min at 4 °C. The concentration of protein in the supernatant was quantified using the Pierce™ BCA protein assay kit. After quantitation, equal amounts of protein were loaded on 12% SDS–PAGE for electrophoresis and transferred to a polyvinylidene fluoride membrane. The membrane was blocked with 5% skim milk in Tris-buffered saline containing 0.5% Tween-20 and incubated overnight at 4 °C with primary antibodies (FLG, LOR, IVL, and β-actin) at a dilution of 1:1000. The following day, the membranes were treated with an HRP secondary antibody for 2 h at a dilution of 1:5000 and visualized using a ChemiDoc™XRS + System (Bio-Rad, Richmond, CA, USA). The expression level of each protein was analyzed using Image Lab statistical software (Bio-Rad, CA, USA) and normalized to the expression of β-actin.

### 4.12. Statistical Analysis

Data are presented as the means ± standard deviation (SD). Statistical analysis was performed using one-way analysis of variance (ANOVA) and Tukey’s honestly significant difference test. Statistically significant differences were evaluated using SPSS (SPSS Inc., Chicago, IL, USA).

## Figures and Tables

**Figure 1 ijms-23-00226-f001:**
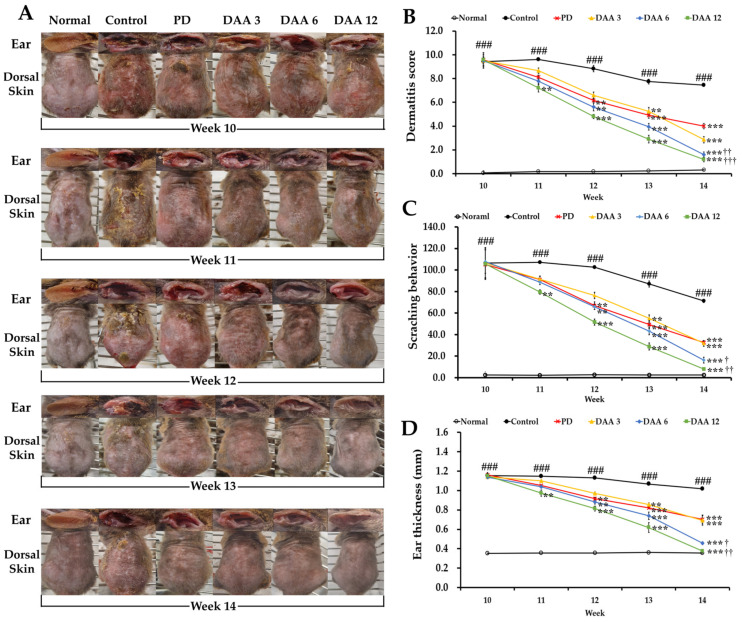
Effects of deacetylasperulosidic acid (DAA) on atopic dermatitis (AD)-like clinical symptoms in 2,4-dinitrochlorobenzene (DNCB)-induced NC/Nga mice. (**A**) Images of the right ear and dorsal skin of mice in each group from weeks 10–14. (**B**–**D**) The clinical features of AD in DNCB-induced NC/Nga mice, that is, (**B**) dermatitis score, (**C**) scratching behavior, and (**D**) ear thickness were evaluated thrice a week during administration of carboxymethyl cellulose (CMC), prednisolone (PD), and DAA. The results were expressed as the mean ± standard deviation (*n = 6*). ### *p* < 0.001 vs. normal (naïve control group); ** *p* < 0.01 and *** *p* < 0.001 vs. control (negative control; DNCB-treated group), PD (positive control; DNCB + prednisolone 3 mg/kg), and DAA (DNCB + DAA 3, 6, or 12 mg/kg) treatment groups; and ^†^
*p* < 0.05, ^††^
*p* < 0.01, and ^†††^
*p* < 0.001 vs. PD treatment group.

**Figure 2 ijms-23-00226-f002:**
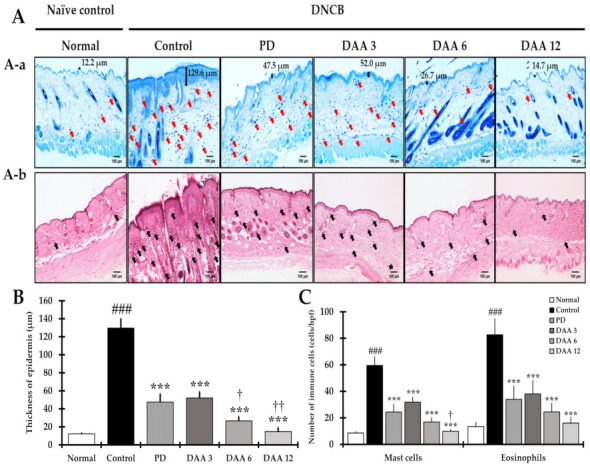
Effects of DAA on DNCB-induced AD-like histopathological alterations in NC/Nga mice. (**A**) Representative images of toluidine blue (TB) and hematoxylin and eosin (H&E) staining in each group. TB or H&E-stained sections were observed under a microscope at 400× magnification (Scale bar = 100 μm). (A-a) Images show TB staining and red arrows indicate mast cells. In addition, the numerical values of the epidermis thickness are indicated on the top. (A-b) Images show H&E staining and black arrows indicate eosinophils. (**B**) The graph shows the epidermal thickness of each group. (**C**) The graph was displayed by measuring the number of infiltrated inflammatory cells (mast cells and eosinophils) in each group. Histological analysis and the number of inflammatory cells were examined in six randomized dorsal skin sites from each mouse. The results are expressed as the mean ± standard deviation (*n = 6*). ### *p* < 0.001 vs. normal (naïve control group); *** *p* < 0.001 vs. control (negative control; DNCB-treated group), PD (positive control; DNCB + prednisolone 3 mg/kg), and DAA (DNCB + DAA 3, 6, or 12 mg/kg) treatment groups; and ^†^
*p* < 0.05 and ^††^
*p* < 0.01 vs. PD treatment group.

**Figure 3 ijms-23-00226-f003:**
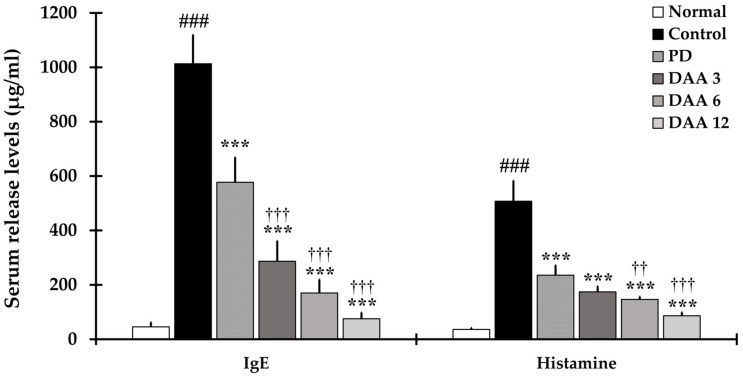
Effects of DAA on serum IgE and histamine levels. Serum IgE and histamine levels were measured using enzyme-linked immunosorbent assay (ELISA). The results were expressed as the mean ± standard deviation (*n = 6*). ### *p* < 0.001 vs. normal (naïve control group); *** *p* < 0.001 vs. control (negative control; DNCB-treated group), PD (positive control; DNCB + prednisolone 3 mg/kg), and DAA (DNCB + DAA 3, 6, or 12 mg/kg) treatment groups; and ^††^
*p* < 0.01 and ^†††^
*p* < 0.001 vs. PD treatment group.

**Figure 4 ijms-23-00226-f004:**
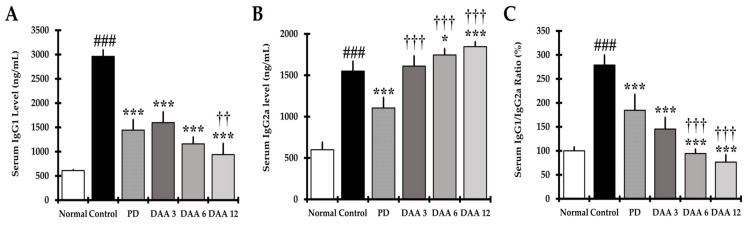
Effects of DAA on IgG1, IgG2a, and IgG1/IgG2a ratios in the serum of DNCB-induced NC/Nga mice. (**A**) IgG1 and (**B**) IgG2a in mouse serum were measured using ELISA. (**C**) Graph represents the ratio of IgG1/IgG2a.The results were expressed as the mean ± standard deviation (*n* = 6). ### *p* < 0.001 vs. normal (naïve control group); * *p* < 0.05 and *** *p* < 0.001 vs. control (negative control; DNCB-treated group), PD (positive control; DNCB + prednisolone 3 mg/kg), and DAA (DNCB + DAA 3, 6, or 12 mg/kg) treatment groups; and ^††^
*p* < 0.01 and ^†††^
*p* < 0.001 vs. PD treatment group.

**Figure 5 ijms-23-00226-f005:**
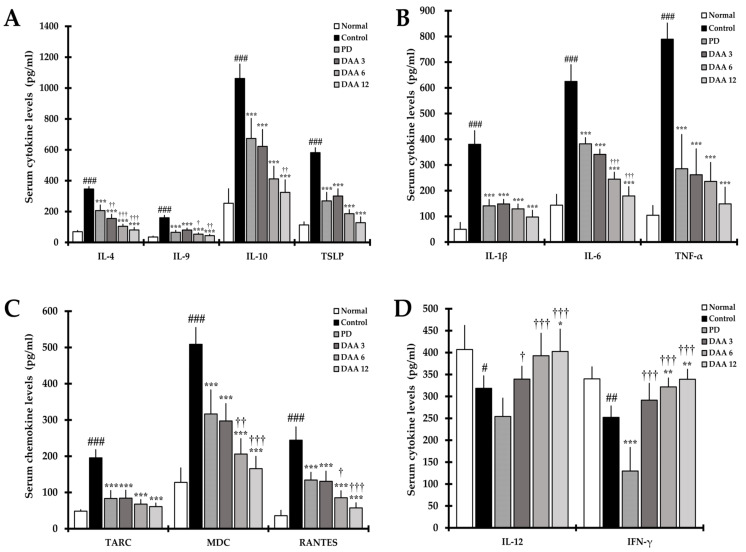
Effects of DAA on AD-associated serum cytokine and chemokine levels in DNCB-induced NC/Nga mice. (**A**) The levels of cytokines interleukin-4 (IL-4), IL-9, IL-10, and thymic stromal lymphopoietin (TSLP) in the serum. (**B**) The levels of cytokines IL-1β, IL-6, and tumor necrosis factor-α (TNF-α) in the serum. (**C**) The levels of the chemokines thymus and activation-regulated chemokine (TARC), macrophage-derived chemokines (MDC), and thymus and activation-regulated chemokine (TARC), regulated on activation, normal T cell expressed and secreted (RANTES) in the serum. (**D**) The levels of the cytokines IL-12 and IFN-γ in the serum. Cytokine and chemokine levels in mouse serum were measured using ELISA. The results were expressed as the mean ± standard deviation (*n* = 6). # *p* < 0.05, ## *p* < 0.01, and ### *p* < 0.001 vs. normal (naïve control group); * *p* < 0.05, ** *p* < 0.01, and *** *p* < 0.001 vs. control (negative control; DNCB-treated group), PD (positive control; DNCB + prednisolone 3 mg/kg), and DAA (DNCB + DAA 3, 6, or 12 mg/kg) treatment groups; and ^†^
*p* < 0.05, ^††^
*p* < 0.01, and ^†††^
*p* < 0.001 vs. PD treatment group.

**Figure 6 ijms-23-00226-f006:**
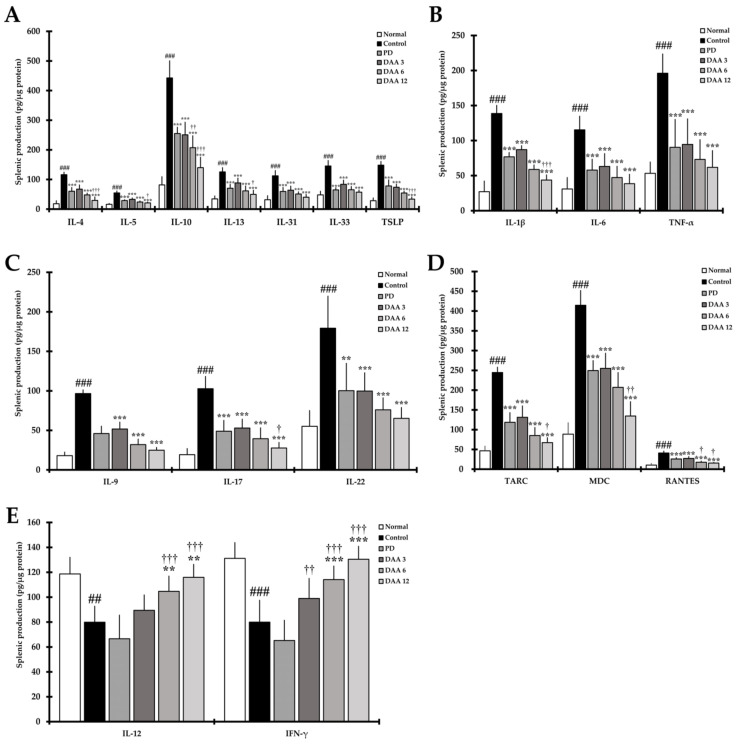
Effects of DAA on the regulation of AD-related cytokines and chemokines secreted from splenocytes of DNCB-induced NC/Nga mice. (**A**) The levels of cytokines IL-4, IL-5, IL-10, IL-13, IL-31, IL-33, and TSLP in the supernatant of splenocytes. (**B**) The levels of the cytokines IL-1β, IL-6, and TNF-α in the supernatant of splenocytes. (**C**) The levels of the cytokines IL-9, IL-17, and IL-22 in the supernatant of splenocytes. (**D**) The levels of the chemokines TARC, MDC, and RANTES in the supernatant of splenocytes. (**E**) The levels of the cytokines IL-12 and IFN-γ in the supernatant of splenocytes. Splenocytes were stimulated with Con-A for 72 h, and cytokines and chemokines levels in the supernatant were investigated using ELISA. Cytokines were normalized to the protein concentration of the lysate. The results were expressed as the mean ± standard deviation (*n* = 6). ## *p* < 0.01 and ### *p* < 0.001 vs. normal (naïve control group); ** *p* < 0.01 and *** *p* < 0.001 vs. control (negative control; DNCB-treated group), PD (positive control; DNCB + prednisolone 3 mg/kg), and DAA (DNCB + DAA 3, 6, or 12 mg/kg) treatment groups; and ^†^
*p* < 0.05, ^††^
*p* < 0.01, and ^†††^
*p* < 0.001 vs. PD treatment group.

**Figure 7 ijms-23-00226-f007:**
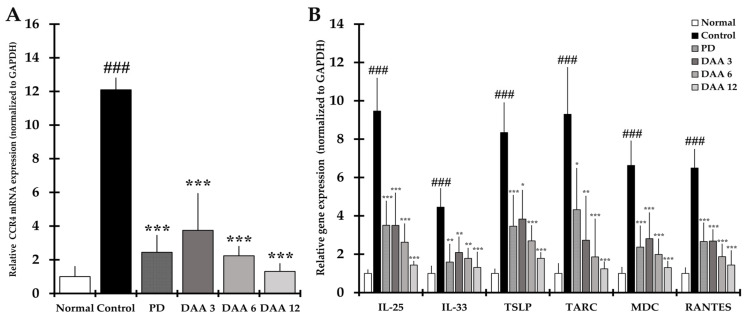
Effects of DAA on gene expression of cytokines, chemokines, and CCR4 associated with the activation of Th2 cells in the dorsal skin of DNCB-induced NC/Nga mice. (**A**) CCR4 gene expression level in the dorsal skin. (**B**) IL-25, IL-33, TSLP, TARC, MDC, and RANTES gene expression level in the dorsal skin. The mRNA levels of all genes were normalized to those of GAPDH. Results were expressed as the mean ± standard deviation (*n* = 6). ### *p* < 0.001 vs. normal (naïve control group) and * *p* < 0.05, ** *p* < 0.01, and *** *p* < 0.001 vs. control (negative control; DNCB-treated group), PD (positive control; DNCB + prednisolone 3 mg/kg), and DAA (DNCB + DAA 3, 6, or 12 mg/kg) treatment groups.

**Figure 8 ijms-23-00226-f008:**
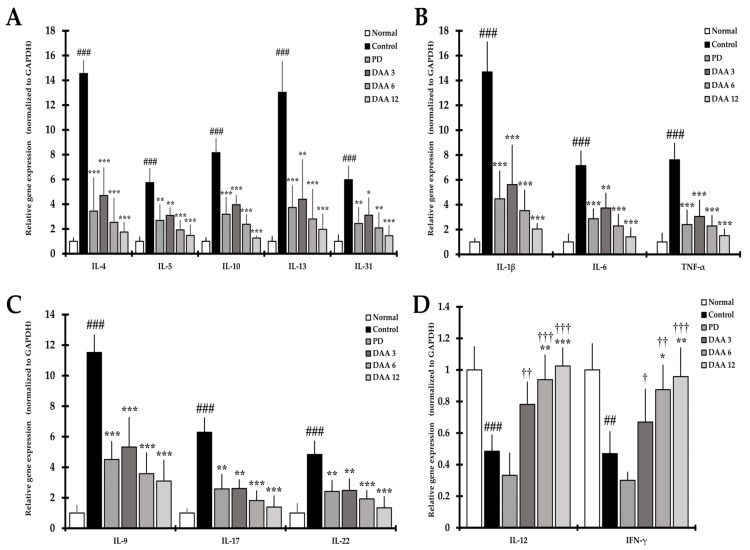
Effects of DAA on the regulation of gene expression of AD-related cytokines and chemokines in the dorsal skin of DNCB-induced NC/Nga mice. (**A**) IL-4, IL-5, IL-10, IL-13, and IL-31 gene expression level in the dorsal skin. (**B**) IL-1β, IL-6, and TNF-α gene expression level in the dorsal skin. (**C**) IL-9, IL-17, and IL-22 gene expression level in the dorsal skin. (**D**) IL-12 and IFN-γ gene expression level in the dorsal skin. The mRNA levels of all genes were normalized to those of GAPDH. The results were expressed as the mean ± standard deviation (*n* = 6). ## *p* < 0.01, and ### *p* < 0.001 vs. normal (naïve control group); * *p* < 0.05, ** *p* < 0.01, and *** *p* < 0.001 vs. control (negative control; DNCB-treated group), PD (positive control; DNCB + prednisolone 3 mg/kg), and DAA (DNCB + DAA 3, 6, or 12 mg/kg) treatment groups; and ^†^
*p* < 0.05, ^††^
*p* < 0.01, and ^†††^
*p* < 0.001 vs. PD treatment group.

**Figure 9 ijms-23-00226-f009:**
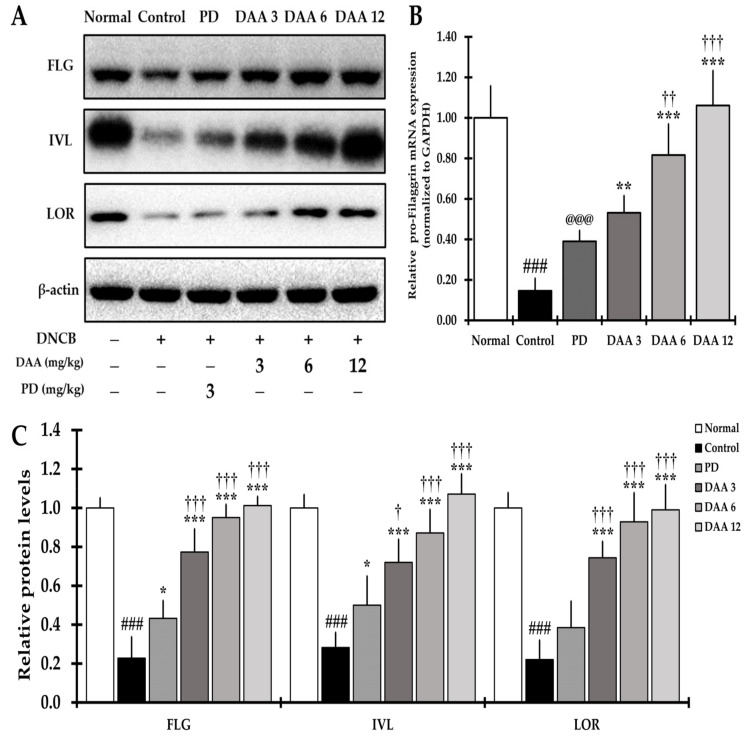
Effects of DAA on the restoration of skin barrier proteins in the dorsal skin of DNCB-induced NC/Nga mice. (**A**) Density levels of skin barrier-associated proteins in the dorsal skin of NC/Nga mice. (**B**) pro-FLG gene expression level in the dorsal skin. The level of pro-FLG mRNA was normalized to that of GAPDH. (**C**) Relative protein expression of FLG (34 kDa), IVL (68 kDa), and LOR (26 kDa) normalized against protein expression of β-actin (43 kDa). The results are expressed as the mean ± standard deviation (*n* = 6). ### *p* < 0.001 vs. normal (naïve control group), * *p* < 0.05, ** *p* < 0.01, and *** *p* < 0.001 vs. control (negative control; DNCB-treated group), PD (positive control; DNCB + prednisolone 3 mg/kg), and DAA (DNCB + DAA 3, 6, or 12 mg/kg) treatment groups, ^†^
*p* < 0.05, ^††^
*p* < 0.01 and ^†††^
*p* < 0.001 vs. PD treatment group, and ^@@@^
*p* < 0.001 vs. normal (naïve control group). FLG, filaggrin; IVL, involucrin; LOR, loricrin; pro-FLG, pro-filaggrin.

**Figure 10 ijms-23-00226-f010:**
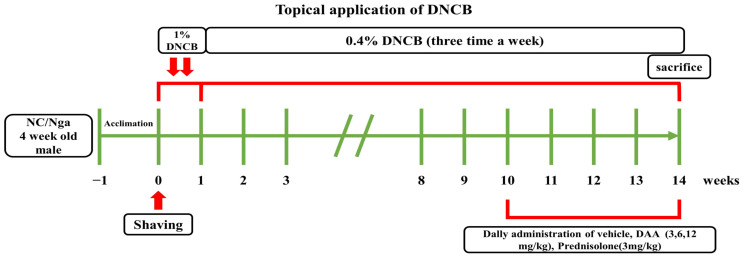
Schematic diagram of DNCB treatment and oral administration of DAA. Four-week-old male NC/Nga mice were acclimated for 1 week, and the hair on the dorsal skin was shaved. Thereafter, 1% DNCB was applied twice to the dorsal skin and right ear. Thereafter, 0.4% DNCB was applied thrice a week until the 14th week. DAA and PD were orally administered from weeks 10–14.

**Table 1 ijms-23-00226-t001:** Primer sequences (in vivo).

Gene	Forward	Reverse
IL-1β (m)	TGT GTT TTC CTC CTT GCC TCT GAT	TGC TGC CTA ATG TCC CCT TGA AT
IL-4 (m)	ACG GAG ATG GAT GTG CCA AAC	AGC ACC TTG GAA GCC CTA CAG A
IL-5 (m)	TCA GCT GTG TCT GGG CCA CT	TT ATG AGT AGG GAC AGG AAG CCT CA
IL-6 (m)	CCA CTT CAC AAG TCG GAG GCT TA	GCA AGT GCA TCA TCG TTG TTC ATA C
IL-9 (m)	GGG CAT CAG AGA CAC CAA T	GGA CGG AGA GAC ACA AGC A
IL-10 (m)	GAC CAG CTG GAC AAC ATA CTG CTA A	GAT AAG GCT TGG CAA CCC AAG TAA
IL-12 (m)	TGA ACT GGC GTT GGA AGC	GCG GGT CTG GTT TGA TGA
IL-13 (m)	CAA TTG CAA TGC CAT CTA CAG GAC	CGA AAC AGT TGC TTT GTG TAG CTG A
IL-17 (m)	AAG GCA GCA GCG ATC ATC C	GGA ACG GTT GAG GTA GTC TGA G
IL-22 (m)	CAG CTC CTG TCA CAT CAG CGG T	AGG TCC AGT TCC CCA ATC GCC T
IL-25 (m)	CTC AAC AGC AGG GCC ACT C	GTC TGT AGG CTG ACG CAG TGT G
IL-31 (m)	ATA CAG CTG CCG TGT TTC AG	AGC CAT CTT ATC ACC CAA GAA
IL-33 (m)	GAT GAG ATG TCT CGG CTG CTT G	AGC CGT TAC GGA TAT GGT GGT C
IFN-γ (m)	CGG CAC AGT CAT TGA AAG CCT A	GGC ACC ACT AGT TGG TTG TCT TTG
TNF-α (m)	TAC TGA ACT TCG GGG TGA TTG GTC	CAG CCT TGT CCC TTG AAG AGA ACC
TSLP (m)	TGC AAG TAC TAG TAC GGA TGG GGC	GGA CTT CTT GTG CCA TTT CCT GAG
CCR4 (m)	TCT ACA GCG GCA TCT TCT TCA T	CAG TAC GTG TGG TGG TGC TCT G
Pro-Filaggrin (m)	GAA TCC ATA TTT ACA GCA AAG CAC CTT G	GGT ATG TCC AAT GTG ATT GCA CGA TTG
GAPDH (m)	ACT TTG TCA AGC TCA TTT CC	TGC AGC GAA CTT TAT TGA TG

## Data Availability

The data presented in this study are available upon request from the corresponding author.

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
