# Peer review of "Deacetylasperulosidic Acid Ameliorates Pruritus, Immune Imbalance, and Skin Barrier Dysfunction in 2,4-Dinitrochlorobenzene-Induced Atopic Dermatitis NC/Nga Mice"

_ijms, 2021, doi:10.3390/ijms23010226_

Round 1
Reviewer 1 Report
The study presented by the authors aims to validate the efficacy of DAA in the treatment of atopic dermatitis in an animal model. They evaluated the goodness of the nutraceutical by dosing severe interleukins and skin proteins. Moreover they compared results with cortisone administration. All the results presented confirmed the efficacy of DAA both on markers and on skin.
the paper is clear.
Author Response
Thank you very much for taking the time out of your busy schedule to review our thesis. We believe that our paper will provide insight into atopic dermatitis in the future.
kind regards,
Reviewer 2 Report
The study led by Su Oh et al., has systematically explored the therapeutic potential of Deacetylasperulosidic acid (DAA) towards atopic dermatitis using a mouse model of AD. The authors have elaborately measured the various Th2 cytokines and Th2 activating cytokine levels in control and DAA treated mice to demonstrate its potential. It is indeed fascinating study that highlights on the potential of DAA in alleviating AD symptoms and pathology. However, there are concerns listed below that needs to be addressed to strengthen this study.
Major
- Authors have used only DNCB based AD model using NC/Nga mice. However, to demonstrate its diverse potential in limiting or alleviating AD, they need to use at least one more AD model to demonstrate its therapeutic potential such as ovalbumin or mc903 based models.
- In Figure 6, the authors have checked the levels of IL13 in splenocytes obtained from the mice treated with DAA and DNCB (NC/Nga). However, in Figure 5, authors have not provided the levels of IL13. Did the authors check the IL13 levels in the serum samples of the mice treated with DAA+DNCB (NC/Nga)?
- In line 273, Page.10, the authors have mentioned that the PD did not show an increase in the expression of skin barrier proteins FLG, IVL, and LOR. However, the data on Figure 9A and C shows that it does show an effect to increase. The authors can correct or explain on this. And in 9B, the statistical comparison between PD and normal is missing.
- IL10 is anti-inflammatory. Can the authors comment on its increase in the serum of control mice (Figure 6A) while in the dorsal skin the expression of IL10 is only secondary to IL4 and IL13 (as expected).
Minor
- Figure 1A needs to be labelled for ear and dorsal side pictures and ideally a line of gap between them for the easy identification of the display of these two sample types within a single figure.
- The first line of the abstract may needs to be re-written as it appears not clear.
Author Response
Thank you very much for taking the time out of your busy schedule to review our thesis.
We believe that our paper will provide insight into atopic dermatitis in the future.
We have revised the paper at your request.
Please refer to the attached file.

Round 2
Reviewer 2 Report
Thank you for addressing the concerns. I agree to the responses provided by the authors. Now, I recommend the acceptance of the publication with the following minor correction in the figures.
Please increase the font size of the Y axis labeling in all the figures to improve on the readability of the publication.
Author Response
Thank you again for reviewing our thesis.
Following the reviewer's recommendations, we reviewed the font size of the figures.
Also, depending on the situation, the size of the x-axis was also adjusted.
In the case of Figure 1, the size has been adjusted to make it easier to distinguish the "dermatitis score", "ear thickness", and "scratching behavior" items. (B-D)
Figure 1 had a feeling of being stretched sideways in the previous manuscript.
We have corrected that part this time.
For the reader's readability, the size of the figure in the manuscript has also been modified appropriately.
Thank you very much for reviewing again.
The revised manuscript is attached.
kind regards,
